# Enhancing *in vitro* ruminal digestibility of oil palm empty fruit bunch by biological pre-treatment with *Ganoderma lucidum* fungal culture

F. M. Y. Nur-Nazratul[1], M. R. M. Rakib[1]*, M. Z. Zailan[1‡], H. Yaakub[2‡]

**1** Faculty of Sustainable Agriculture, Universiti Malaysia Sabah, Sandakan, Sabah, Malaysia, **2** Department of Animal Science, Faculty of Agriculture, Universiti Putra Malaysia, Serdang, Selangor, Malaysia

☯ These authors contributed equally to this work.
‡ These authors also contributed equally to this work.
* rakibmrm@ums.edu.my

**Data Availability Statement:** All relevant data are within the manuscript and its Supporting Information files.

## Abstract

The changes in lignocellulosic biomass composition and *in vitro* rumen digestibility of oil palm empty fruit bunch (OPEFB) after pre-treatment with the fungus *Ganoderma lucidum* were evaluated. The results demonstrated that the pre-treatment for 2–12 weeks has gradually degraded the OPEFB in a time-dependent manner; whereby lignin, cellulose, and hemicellulose were respectively degraded by 41.0, 20.5, and 26.7% at the end of the incubation period. The findings were corroborated using the physical examination of the OPEFB by scanning electron microscopy. Moreover, the OPEFB pre-treated for 12 weeks has shown the highest *in vitro* digestibility of dry (77.20%) and organic (69.78%) matter, where they were enhanced by 104.07 and 96.29%, respectively, as compared to the untreated control. The enhancement in the *in vitro* ruminal digestibility was negatively correlated with the lignin content in the OPEFB. Therefore, biologically delignified OPEFB with *G. lucidum* fungal culture pre-treatment have the potential to be utilized as one of the ingredients for the development of a novel ruminant forage.

## Introduction

The livestock industry is one of Malaysia's most important and fundamental industries as it contributes significantly to agricultural development and food security. It supplies domestic requirements, including eggs, milk, dairy products, and meats, in which the last-mentioned is the prominent source of animal proteins in the diet of the Malaysian population [1]. This industry comprises two different sectors, i.e., non-ruminant and ruminant [2]. Since 2017, the self-sufficiency level (SSL) of non-ruminant (pork, poultry meat, and eggs) has been increased, ranging from 90–118% in 2019. Conversely, the SSL of beef and mutton in the same period was only 22 and 10%, respectively, and it has never reached more than 25% since 2014 [3]. These statistics indicated that ruminant products in Malaysia are still inadequate to meet the

**Funding:** This project was funded by the Ministry of Higher Education, Malaysia (https://mohe.gov.my/en/) under the Fundamental Research Grant Scheme (Grant No.: FRG0478-2017). The fund was awarded to MRMR. The funders had no role in study design, data collection and analysis, decision to publish, or preparation of the manuscript.

**Competing interests:** The authors have declared that no competing interests exist.

domestic demand, and the situation is getting alarming due to an uninterrupted increase in human population and consumption. Consequently, Malaysia has to rely heavily on importation, especially from India, Australia, and New Zealand to satisfy the growing demand, leading to an elevation in the trade balance of food products [4].

Several factors have contributed to the ruminant sector's lag, such as lack of land area suitable for grazing, irregular supply of feed with high nutritive value, high cost of feed, low supply of quality breeding stock, and inefficient marketing system [4]. In this regard, many strategies have been introduced to enhance the efficiency of ruminant production. One of them is through increasing local feed production by utilising agricultural by-products, either from the oil palm or rice industry [5].

At present, Malaysia is one of the world's largest palm oil producers. In 2020, Malaysia produced 19.9 million tonnes of palm oil, which contributed to 26% of the global palm oil supply [6]. During extraction of palm oil in a mill, it generates several by-products, including the oil palm empty fruit bunches (OPEFB), which is a lignocellulosic biomass waste. On average, for every tonne of fresh fruit bunch, 230 to 250 kg (23–25%) of OPEFB is generated [7], and it was reported that more than 19 million tonnes of OPEFB are produced annually [8]. The OPEFB have a high abundance but low commercial value and decomposition rate, hence causing disposal problems and environmental pollution [9]. Therefore, putting OPEFB to good use in the agricultural sector may provide a better solution for sustainable agricultural system and pollution issues.

The OPEFB could be one of the roughage sources for ruminants as it comprises a high amount of structural carbohydrates, such as cellulose and hemicellulose [10]. Nevertheless, the potential of OPEFB as a ruminant feed is still constrained by its low digestibility, as it naturally contains a resistant carbohydrate–lignin shield (19–30% of total mass) that is difficult to be degraded through anaerobic fermentation in the rumen [11]. The presence of lignin on OPEFB impeded the accessibility of digestive enzymes to the cellulose and hemicellulose [12]. Thus, the OPEFB needs pre-treatment to break down the lignin shield (delignification) and disrupt the crystalline structure of cellulose for better digestion by ruminants.

Several pre-treatments have been investigated to improve the feeding quality of forages, including physical, chemical, and biological. Nonetheless, the application of physical and chemical pre-treatments has various limitations, as they are not economical, producing potential pollutants, and possessing high environmental risk. Alternatively, biological pre-treatment using microbes and their enzyme extracts for delignification has been extensively studied, and it is regarded as more economical and environmentally friendly [13].

White-rot fungi (WRF) have been exploited to improve the nutritive value and digestibility of many potential agricultural by-products for ruminant nutrition [12, 14]. They are wood-decaying basidiomycetes which suggested to be the most effective lignin-degrading microbes in nature, as they are capable of secreting three different ligninolytic enzymes, viz., lignin peroxidase (LiP), manganese peroxidase (MnP), and laccase (Lac) [15]. WRF such as *Lentinula edodes*, *Pleurotus eryngii*, *Pleurotus ostreatus*, *Phanerochaete chrysosporium*, and *Tramates versicolor* have been reported as selective lignin-degrading fungi and are efficient in improving the degradability of by-products such as corn stove, wheat straw, and rice straw [14, 16]. The fact that different fungal species possess different selectivity in degrading lignin, it is important to identify a specific combination of a fungus with a substrate for the highest result of delignification.

The fungus *Ganoderma lucidum*, commonly known as *Lingzhi* or *Reishi* has been used as a health-promoting and therapeutic agent for more than 2000 years [17, 18]. It was easily found in local and around Asia, where it grows in the tropics with various hot and humid locations [19]. Apart from its high medicinal values, it was reported as a specific lignin-degrading fungus

and has improved *in vitro* digestibility of many lignocellulosic materials in rumen fluid [16]. In addition, *G. lucidum* has also been well exploited as a poultry feed supplement to diverse medical conditions. As such, the application of antibiotics to poultry can be minimised since this fungal supplement contains essential nutrients and beneficial bioactive compounds [20]. Despite these, the application of *G. lucidum* in the biological pre-treatment of OPEFB remains elusive.

Herein, we evaluated the changes in fibre composition and *in vitro* rumen digestibility of OPEFB pre-treated with *G. lucidum* fungal culture across different incubation period. In the first instance, the fibre composition of OPEFB was analysed. Furthermore, the physical changes of the OPEFB after pre-treatment with *G. lucidum* were examined using scanning electron microscopy (SEM). Lastly, the *in vitro* rumen gas production and digestibility of the pre-treated OPEFB were determined.

## Materials and methods

### Preparation of pure *G. lucidum* fungal culture

*Ganoderma lucidum* strain IUM 4303 (JQ520182.1) was obtained from a local mushroom farm in Sabah, Malaysia (5˚ 55.274' N, 116˚ 9.156' E). Its pure culture was obtained from the basidiocarp and isolated from potato dextrose agar (PDA) plates using standard tissue culture technique. Stock culture (slant) was then prepared and maintained at 4˚C in PDA for further use, while working culture was prepared by sub-culture and incubated for seven days at 25–28˚C under dark condition [21].

### Preparation of OPEFB and fungal culture pre-treatment

The shredded OPEFB with an average length of approximately 5 cm was obtained from a local palm oil mill located in Sabah, Malaysia (5˚ 5.086' N, 118˚ 35.307' E). It was sun-dried to prevent the growth of moulds, weighed for 100 g, and then soaked in distilled water overnight. Afterwards, the water was drained out, and the OPEFB was filled into 9 × 35 cm polypropylene bags and sterilised using an autoclave machine (Tomy, SX-700) at 121˚C and 15 p.s.i. pressure for 15 minutes. The sterilised OPEFB was left to cool at room temperature (25–28˚C) prior to the pre-treatment with *G. lucidum*.

The sterilised OPEFB was pre-treated (inoculated) with seven-day-old active *G. lucidum* pure culture at the rate of 50 mg kg$^{-1}$ (dry weight). The pre-treated OPEFB was incubated in room condition at 25–28˚C and 80–85% relative humidity in the dark for 2, 4, 6, 8, 10, and 12 weeks. Untreated (uninoculated) OPEFB served as a control. The experiment was conducted with four replications for each treatment period, resulting in a total of 24 experimental units, excluding the control. All experimental units were performed in a completely randomised design (CRD). After each treatment period, all four bags of the pre-treated OPEFB were harvested, homogenised, and oven-dried at 70˚C until constant weight is obtained. The dried samples were cut into smaller sections and kept in sealed bags for further analysis. The same procedures were applied on untreated OPEFB samples.

### Fibre composition analysis

Lignin, cellulose, hemicellulose, and dry matter contents in the OPEFB were determined gravimetrically, in which the fibre factions, namely neutral detergent fibre (NDF), acid detergent fibre (ADF), and acid detergent lignin (ADL) were analysed as previously described by Van Soest et al. [22]. The ADL represents the lignin content, while the hemicellulose content was calculated as the difference between NDF and ADF, and the difference between ADF and ADL

represents the cellulose content. All samples were analysed in triplicates, and expressed in g kg$^{-1}$ dry matter (DM).

## Scanning electron microscopy (SEM)

The analysis was conducted using a MA10 (Carl ZEISS) scanning electron microscope to observe the physical changes of the OPEFB after pre-treatment with *G. lucidum*. During preparation, one small section (1–5 mm) of the sample from each treatment group was cut and coated with a thin layer of gold using a K550K (Emitech) sputter coater [23].

## *In vitro* rumen gas production and digestibility

*In vitro* gas production of OPEFB sample was determined following the method described by Menke and Steingass [24], which was used to simulate the rumen anaerobic fermentation activity in ruminants. The experiment was performed in duplicates to ensure the consistency of rumen fluid activity. Rumen fluid was collected from a fistulated cow from the Faculty of Agriculture, Universiti Putra Malaysia (2˚ 59.102' N, 101˚ 44.176' E), which has been fed with 40% concentrate and 60% hay. All facets of animal care and use met the requirements of the Universiti Putra Malaysia Policy and Code of Practice for the Care and Use of Animal for Scientific Purposes. The protocol was approved by the Institutional Animal Care and Use Committee (IACUC) of the Universiti Putra Malaysia (Animal Utilisation Protocol Number: UPM/IACUC/AUP-R063/2020). The rumen fluid was collected before morning feeding, and filtered using a muslin cloth to remove feed particles. The rumen fluid was mixed with the buffer medium in a ratio of 1:2 (v/v), in which the buffer medium contains micromineral (13.2 g $CaCl_2.2H_2O$, 10 g $MnCl_2.4H_2O$, 1 g $CoCl_2.6H_2O$, 8 g $FeCl_2.6H_2O$, in 100 mL of distilled water), buffer (39 g $NaHCO_3$ in 1 L of distilled water), micromineral (5.7 g $Na_2HPO_4$, 6.2 g $KHPO_4$, 0.6 g $MgSO_4.7H_2O$, in 1 L of distilled water), 0.1% resazurin, and reducing solution (4 mL of 1 N NaOH, 625 mg $Na_2S.7H_2O$, in 95 mL of distilled water). The OPEFB samples and standard hays were weighed for 200 mg and put into glass syringes. All samples and standard hays were prepared in triplicates including blank (without sample). The mixture of rumen fluid with buffer medium was pumped for 30 mL, and all syringes were incubated at 39˚C. The volume of gas produced was determined at 0, 2, 4, 6, 8, 10, 12, 24, 36, 48, and 72 hours of the incubation period. The gas production data were fitted to the equation by Orskov and McDonald [25] using a NEWAY computer program. Fermentation kinetics were calculated according to equation $P = a + b\,(1\text{-}e^{-ct})$, where $P$ is the gas volume produced at the time '$t$', $a$ is the gas volume produced from the immediately soluble fraction, $b$ is the gas volume produced from the insoluble fraction, $c$ is the gas production rate constant for the insoluble fraction ($b$), $t$ is the incubation time, and ($a + b$) is the potential extent of gas production.

After 48 hours of incubation, three syringes of samples from each treatment were taken out to determine the *in vitro* dry matter digestibility (IVDMD) and organic matter digestibility (IVOMD). The residues were filtered using a sintered glass crucible (coarse porosity no. 1, pore size 100–160 μm), and then oven-dried at 105˚C for 48 hours to estimate the disappearance of dry matter. The organic matter of dry residue was determined gravimetrically by incineration in a muffled furnace at 550˚C for 3 hours.

## Statistical analysis

One-way analysis of variance (ANOVA) was performed, and the treatment means were compared using Duncan's multiple range test at a significance level of 5% ($p \leq 0.05$). The data were subjected to Pearson's correlation analysis to evaluate the relationship between the variables, and regression analysis was carried out to evaluate the trend across the treatment period,

where the trends were fitted to a linear model. All statistical analysis was performed using a Statistical Analysis System Version 9.4 (SAS 9.4).

# Results and discussion

## Fibre compositions

The OPEFB applied in this study contains 35.77 g kg$^{-1}$ DM of lignin, and the remaining are carbohydrates in the form of cellulose (78.28 g kg$^{-1}$ DM) and hemicellulose (54.41 g kg$^{-1}$ DM). The fibre compositions of OPEFB were changed significantly after pre-treatment with *G. lucidum* in a time-dependent manner (S1 Table and Fig 1). Since the study aims to modify the OPEFB into animal feeds (forages), two common indicators in relation to animal nutrition, namely, NDF and ADF were used to evaluate the fibre content in OPEFB; in which NDF represents the structural component of forages, while ADF specifically represents the least digestible part (e.g., cellulose and lignin) in the forages [26]. The results demonstrated that the reduction of NDF and ADF was negatively correlated with the treatment period. The NDF of the pre-treated OPEFB was decreased from 251.12 g kg$^{-1}$ to 241.90 g kg$^{-1}$ at week six and further reduced to 202.88 g kg$^{-1}$ at week 12 (Fig 1A). Similarly, the ADF after pre-treatment was also decreased gradually from 196.71 g kg$^{-1}$ to 162.99 g kg$^{-1}$ at week 12 (Fig 1B).

Additionally, the change in lignin content was negatively correlated to the treatment period in a linear manner ($R^2$ = 0.9016). In contrast to NDF and ADF, a significant reduction of lignin content in the OPEFB was observed as early as two weeks of pre-treatment and was gradually reduced to 21.11 g kg$^{-1}$ on week 12 (Fig 1C). Similarly, a significant decrease in cellulose and hemicellulose content was recorded after six weeks of pre-treatment. By the end of the pre-treatment, both cellulose and hemicellulose contents were further degraded in a linear trend to 62.24 g kg$^{-1}$ ($R^2$ = 0.8255) and 39.89 g kg$^{-1}$ ($R^2$ = 0.9185), respectively (Fig 1D and 1E). In addition, correlation analysis revealed that the changes in lignin, cellulose, and hemicellulose were significantly correlated with each other ($p < 0.001$), with Pearson's correlation coefficients ranging from 0.68 to 0.78. Nonetheless, further investigations on the rate of change in the fibre compositions based on the linear equations revealed that the cellulose and hemicellulose were degraded at a higher rate of 1.47 and 1.43 g kg$^{-1}$ per week, respectively, as compared to that of lignin (1.07 g kg$^{-1}$ per week) (Fig 1C–1E). Upon comparison with the untreated control, the fibre compositions of OPEFB pre-treated for 12 weeks has demonstrated a higher overall loss in lignin (41.0%) than cellulose (20.5%) and hemicellulose (26.7%). Lastly, the pre-treatment of OPEFB for 12 weeks has significantly increased the dry matter from 79.47 to 111.49 g kg$^{-1}$ of fresh substrate in a linear trend (Fig 1F).

The decrease in cellulose, hemicellulose, and lignin in the pre-treated OPEFB could be related to the ability of *G. lucidum* to solubilise and utilise the carbohydrates as their carbon sources [27]. Bartnicki-Garcia et al. [28] reported that these carbohydrates are essential for the growth of fungi and the fungal cell wall synthesis. Generally, lignin is a highly dense structure and the most recalcitrant to degradation among all fibre compositions. It linked to both hemicellulose and cellulose by forming a physical seal that prevents the penetration of hydrolytic enzymes, thus the components underneath (hemicellulose and cellulose) were inhibited from being released and utilised [29]. In the current study, the biological degradation of the lignin component could be the result of ligninolytic enzymes (LiP, MnP, and Lac) produced by *G. lucidum*. Jalc [30] reported that WRF was capable to selectively degrade lignin, particularly during its early phase of colonisation, followed by a fruiting stage where cellulose and hemicellulose are degraded. This explanation is in agreement with the degradation pattern shown in Fig 1, where the lignin biodegradation of the pre-treated OPEFB started relatively early (week two). Other than the three aforementioned ligninolytic enzymes, peroxidases secreted by *G.*

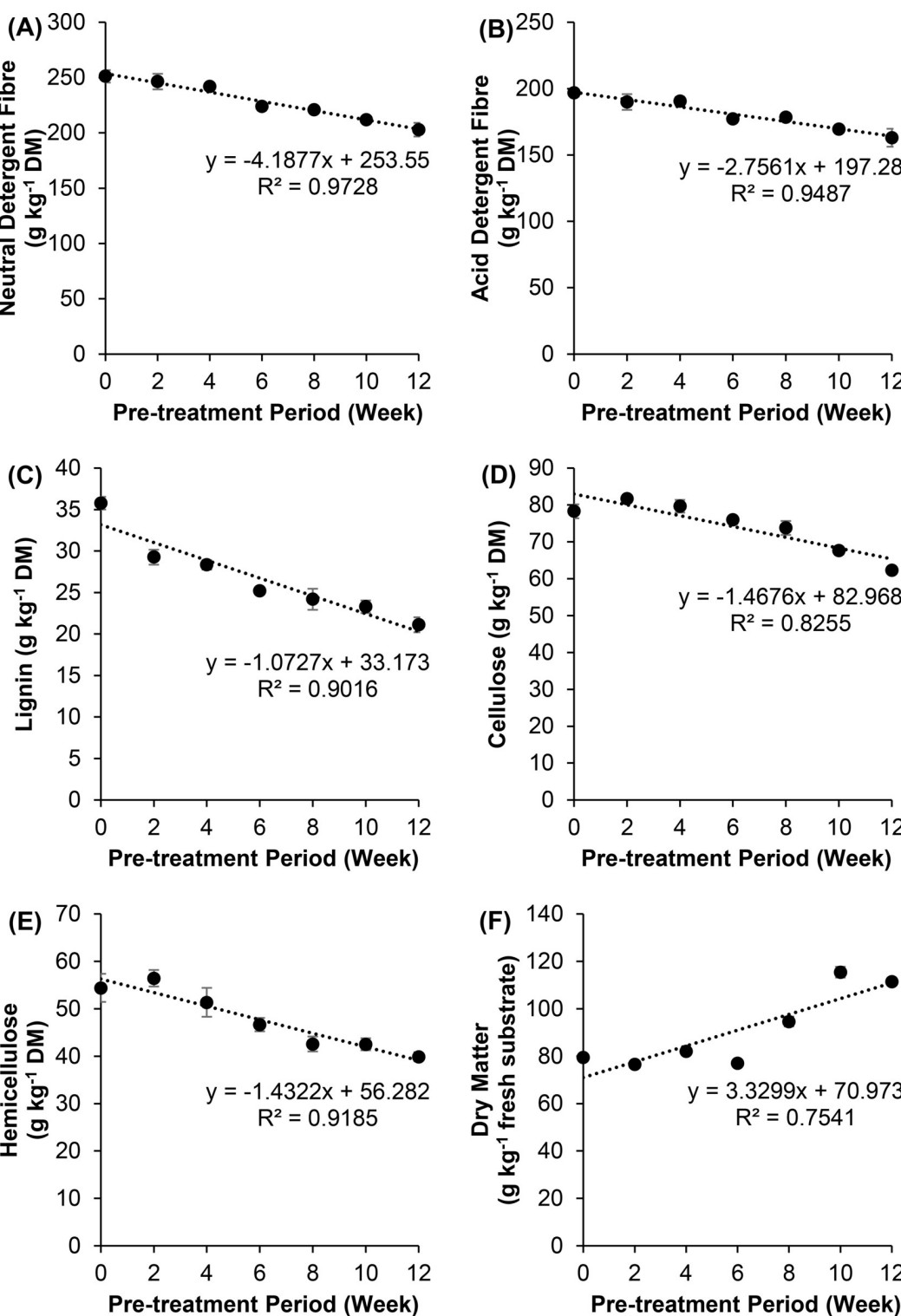

**Fig 1. Changes in fibre composition and dry matter (means ± standard error) of OPEFB pre-treated with *G. lucidum* across treatment period were fitted to linear model.** (A): Neutral detergent fibre. (B): Acid detergent fibre. (C): Lignin. (D): Cellulose. (E): Hemicellulose. (F): Dry matter.

*lucidum* can generate radicals, which are sufficiently small to penetrate and mineralise the dense lignin structure at a distance away from the fungi [31]. In line with several previously reported studies [32, 33], it can be suggested that the biological treatment of OPEFB using *G. lucidum* is capable of disrupting and biologically degrading the lignin-carbohydrate complex.

## Scanning electron microscopy

A qualitative study on the morphological change of pre-treated OPEFB was conducted through SEM analysis, and the images are presented in Fig 2. The surface morphology of untreated OPEFB was rough, with a thick layer of waxy material covering the whole surface strand (Fig 2A). The waxy layer could be the cuticle that presents in most plants to prevent water loss [34]. There were many spiky silica bodies observed being embedded and dispersed randomly on the surface of untreated OPEFB (Fig 2A), which was consistent with the findings reported by Bahrin et al. [35], and Hamzah et al. [36]. The presence of silica bodies on the fibre surface provides strength and reinforcement to the overall fibres by restricting the sliding motion or slippage between the fibres [37]. Altogether, the structure of untreated OPEFB was rigid and solid due to the participation of many sturdy structures, such as lignin, waxes, and silica bodies [23, 36, 37].

Meanwhile, OPEFB having undergone pre-treatment with *G. lucidum* had severe disruption on biomass fibre structure (Fig 2B and 2C). The pre-treated OPEFB showed the removal of the silica bodies from its surface and left only exposed pores (Fig 2B). Hamzah et al. [36] reported that any fibres without silica bodies could easily be decomposed as there was an increase in surface area after the removal, allowing more enzymatic attack by microorganisms. Law et al. [38] also mentioned that the removal of silica bodies exposed the silica crater bottom,

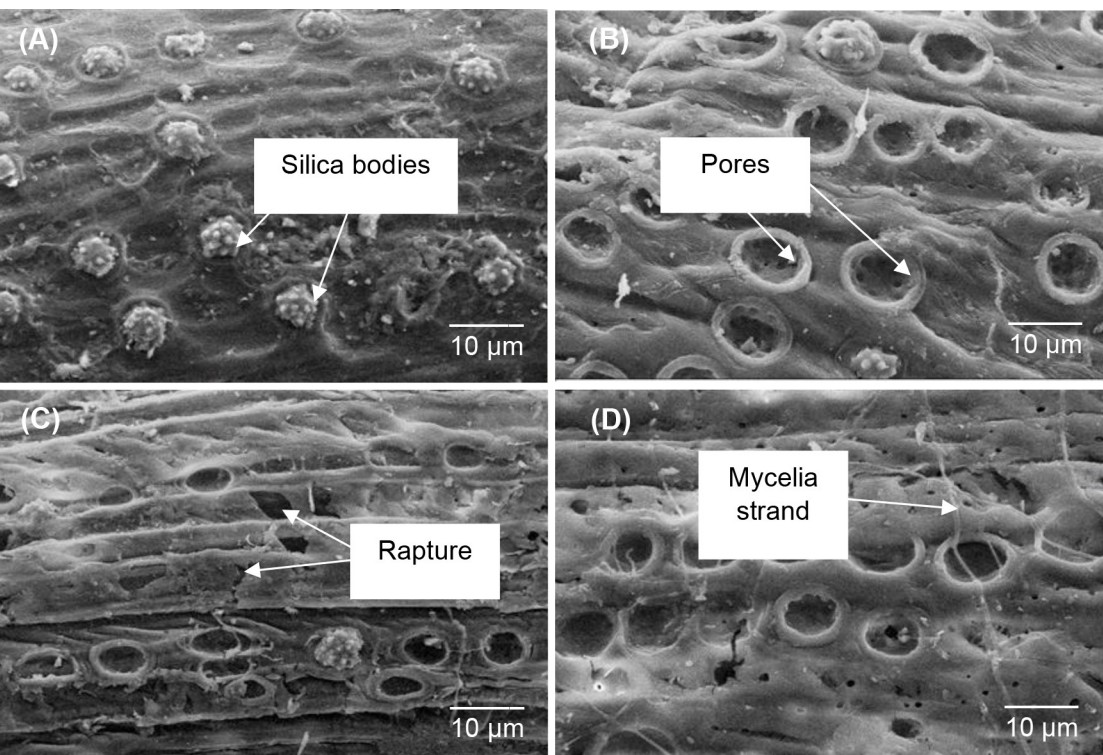

**Fig 2. SEM of OPEFB.** (A): Silica bodies attached on untreated OPEFB. (B): Pores with detached silica bodies on pre-treated OPEFB. (C): Ruptured and rugged surface of pre-treated OPEFB. (D): Mycelia of the *G. lucidum* colonised and penetrated the OPEFB.

which could enhance any chemical or enzymatic penetrations on the fibre surface. Moreover, the removal of these silica bodies has also revealed the siliceous pathway, therefore exposing more amorphous region of the fibres to degradation [39]. Besides, Agbagla-Dohnani et al. [40] reported that silica prevents colonisation by ruminal microorganisms, thereby reducing palatability and degradability of forages in the rumen. Hence, using OPEFB pre-treated with *G. lucidum* as ruminant feed will have a higher surface area and digestibility when consumed by ruminants, as it can be easily attacked by microorganisms present in the rumen.

The surface of pre-treated OPEFB with *G. lucidum* also can be seen ruptured and rugged (Fig 2C), which could be resulted from the hydrolysation of lignin and hemicellulose after the pre-treatment. With reference to Hamzah et al. [36], our fungal pre-treatment has substantially disrupted the fibre structure, increased the surface area, exposed the OPEFB cellulose components, and enhanced the susceptibility of OPEFB to microorganisms in the rumen. Moreover, the surface of pre-treated OPEFB has resulted in a rougher, more wrinkled, and clearer surface, as the layers of matrix materials (e.g., lignin and waxes) were also being removed after the treatment. A study by Kamcharoen et al. [41] supported that after the fungal pre-treatment on OPEFB, the cellulose content increased, leading to a higher digestibility than those of untreated. Isroi et al. [42] also mentioned that fungal treatment on OPEFB caused the structural changes in the lignin by reducing the aromatic units, making the lignin less rigid. Lastly, it was observed that the mycelia of *G. lucidum* have colonised and penetrated the surface of the pre-treated OPEFB (Fig 2D), leading to more porosity and roughness as compared to those untreated, whereby it was in line with the observation reported by Kamcharoen et al. [41] and Isroi et al. [42].

### *In vitro* gas production and digestibility

According to Getachew et al. [43], gas production in an *in vitro* rumen fermentation system is a result of microbial degradation on feed components, particularly carbohydrate fraction. In this regard, any changes in carbohydrate fractions can be reflected by the total gas produced [44]. The cumulative *in vitro* gas production (IVGP) of pre-treated OPEFB with *G. lucidum* at different incubation period (corrected with blank) is shown in S2 Table and Fig 3. Generally, the IVGP of all pre-treated OPEFB increased exponentially as the incubation time increases, and correlation analysis indicated that the IVGP was positively correlated with the pre-treatment period (R = 0.771). It was found that a low IVGP was recorded for the untreated, and pre-treated OPEFB at week two and four. In contrast, the OPEFB pre-treated for 12 weeks showed the highest IVGP, and the effect prolonged until the end of incubation, suggesting that the sample has the highest microbial degradation activity.

Notably, although the OPEFB pre-treated at the first six weeks has a relatively high cellulose and hemicellulose content (Fig 1), they have a lower IVGP and therefore lower degradability than those of pre-treated for 8–12 weeks (Fig 3). This is most probably due to a higher lignin content in OPEFB pre-treated at the first six weeks and this is in line with the correlation analysis, which demonstrated that the IVGP was negatively correlated with the amount of lignin in the OPEFB (R = –0.701). The results were also in accordance with those reported by Sallam et al. [45], which suggested that consuming feed with a lower lignin content has a higher rumen microbial activity as incubation time progress. Altogether, the ensuing pre-treatment of OPEFB has reduced lignin content, which could have released the structural carbohydrate fractions (cellulose and hemicellulose) to be utilised by rumen microbes, therefore causing a higher IVGP.

The calculated kinetics of IVGP is presented in Table 1. It was observed that there was no significant difference for IVGP from a quickly soluble fraction (*a*) at different pre-treatment

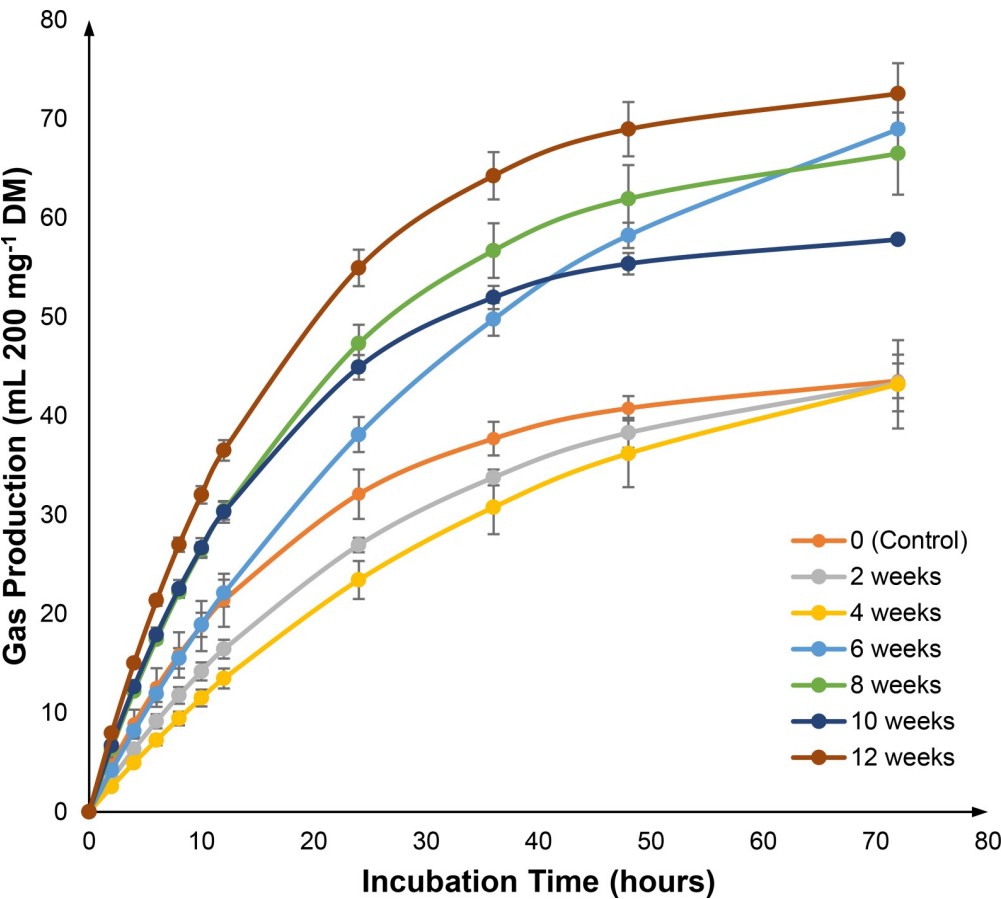

**Fig 3. *In vitro* gas production (mL 200 mg$^{-1}$ dry matter) of OPEFB pre-treated with *G. lucidum* across incubation period.**

period; however, it was significant for IVGP from insoluble (but degradable) fraction (*b*) and their potential gas production (*a* + *b*), as well as the gas production rate (*c*). The IVGP from *b*

**Table 1. *In vitro* gas production kinetics of OPFEB pre-treated with *G. lucidum* at different treatment period.**

| Pre-treatment Period (week) | Gas Production | | | |
|:---:|:---:|:---:|:---:|:---:|
| | (Mean ±Standard error) | | | |
| | *a* | *b* | *a+b* | *c* |
| | (mL) | (mL) | (mL) | (mL hr$^{-1}$) |
| 0 | 1.48 ±1.53 | 46.74 ±1.60c | 45.26 ±2.90e | 0.024 ±0.005a |
| 2 | 0.30 ±0.76 | 47.52 ±4.05c | 47.82 ±4.70de | 0.016 ±0.003bc |
| 4 | 0.15 ±1.66 | 49.22 ±6.03c | 49.37 ±6.37de | 0.012 ±0.000c |
| 6 | 0.91 ±0.46 | 80.90 ±3.37a | 81.81 ±2.95a | 0.012 ±0.001c |
| 8 | -2.28 ±0.43 | 70.89 ±5.03ab | 68.61 ±4.67bc | 0.021 ±0.001ab |
| 10 | -3.22 ±1.26 | 61.81 ±0.30b | 58.59 ±0.96dc | 0.026 ±0.000a |
| 12 | -2.89 ±0.56 | 76.68 ±3.80a | 73.79 ±3.25ab | 0.025 ±0.000a |
| *p* value | 0.0790 | <0.0001 | <0.0001 | 0.0024 |

Data are expressed as the means ± standard errors. Means with the same letter in each column indicates not significantly different ($p \geq 0.05$, Duncan's multiple range test).

was ranged from 46.74 to 80.90 mL, (*a* + *b*) from 45.26 to 81.81 mL, and *c* from 0.012 to 0.026 mL hr$^{-1}$. As expected from Fig 3, the untreated and pre-treated OPEFB at week two and four have low IVGP from *b* and (*a* + *b*), probably due to the high content of lignin, which makes the carbohydrate fraction not accessible for degradation and fermentation by rumen microbes.

Furthermore, it was observed that the pre-treated OPEFB at week six have the highest *b* (80.90 mL) and (*a* + *b*) (81.81 mL), while having the lowest *c* (0.012 mL hr$^{-1}$). In contrast, the pre-treated OPEFB at week 12 has substantial gas production from *b* (76.68 mL) and (*a* + *b*) (73.79 mL), with *c* (0.025 mL hr$^{-1}$) being the highest among most of the others. The results were in accordance with its lowest content of lignin (Fig 1), which makes its carbohydrate fraction most readily available to be degraded and fermented by the rumen microbes.

A further investigation into the digestibility of OPEFB was conducted, and it was found that the IVDMD and IVOMD were changed parallelly, where the *in vitro* digestibility of the pre-treated OPEFB increased significantly beginning from week six. The IVDMD and IVOMD continued to elevate across the treatment period, where at week 12 they have recorded 77.20 and 69.78%, respectively. As compared to the control (week zero), the IVDMD and IVOMD were improved by 104.07 and 96.29%, respectively (S3 Table and Fig 4). The tremendous improvements in the *in vitro* digestibility could be attributed to the substantial biodegradation of lignin in the pre-treated OPEFB. In comparison, the IVDMD and IVOMD reported in the present study were higher than those of Jayanegara et al. [46] which investigated the same lignocellulosic biomass, and comparable with Anassori et al. [47] and Zailan et al. [48] which studied other forages.

Therefore, it was found that the biological delignification of the OPEFB via *G. lucidum* pretreatment can enhance the *in vitro* ruminal digestibility of the material. However, there are a

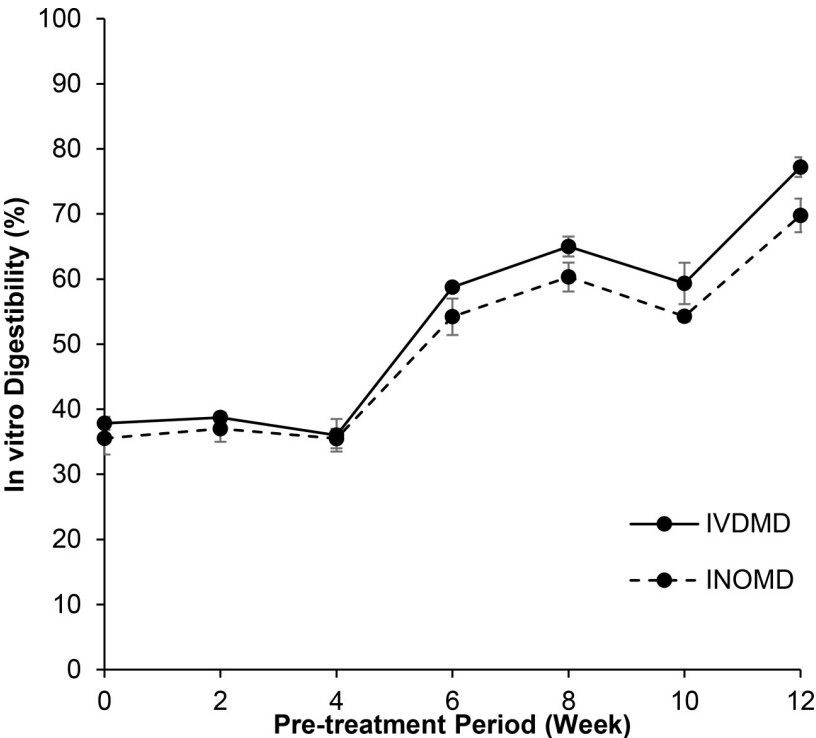

**Fig 4. *In vitro* dry matter digestibility (IVDMD) and organic matter digestibility (IVOMD) of OPEFB pre-treated with *G. lucidum* at 48 hours of *in vitro* gas production.**

few aspects that should be taken into consideration for further development of the biomass into ruminant forages. For instance, it is crucial to identify a solution to upscale the production and optimise the operating conditions to ensure the high efficiency of biomass processing. The delignification mechanism of *G. lucidum* in OPEFB should be clarified to determine its rate-limiting step, which could enhance the efficacy of the process. In addition, an in vivo feeding trial needs to be conducted to evaluate the biomass's palatability, hence its effect on ruminal quality and fermentation characteristics. It has been reported that fungi-treated OPEFB or any other lignocellulosic biomass contains very little crude protein, lipids, vitamins, and minerals [49, 50]. Therefore, it is necessary to incorporate the treated OPEFB with other nutrients as supplements for formulation of a balanced ration. Lastly, the cost-benefit analysis should also be evaluated.

## Conclusions

In summary, pre-treatment of OPEFB with *G. lucidum* has resulted in the gradual degradation of lignin, cellulose, and hemicellulose, where they were respectively degraded by 41.0%, 20.5%, and 26.7% at the end of the incubation period (week 12). In addition to the substantial change in fibre composition, the physical examination of the pre-treated OPEFB has suggested that delignification, removal of silica bodies, and mycelia colonisation were taken place. Moreover, the pre-treatment has improved the *in vitro* ruminal digestibility of OPEFB, where the IVDMD and IVOMD were enhanced by 104.07% and 96.29%, respectively, as compared to the untreated control. Biological delignified OPEFB via *G. lucidum* fungal culture pre-treatment have the potential to be utilized as one of the ingredients for the development of a novel ruminant forage.

## Supporting information

**S1 Table. Changes in fibre composition and dry matter of OPEFB pre-treated with *G. lucidum* across treatment period were fitted to linear model.**
(DOCX)

**S2 Table. *In vitro* gas production (mL 200 mg$^{-1}$ dry matter) of OPEFB pre-treated with *G. lucidum* across incubation period.**
(DOCX)

**S3 Table. *In vitro* dry matter and organic matter digestibility of OPEFB pre-treated with *G. lucidum* at 48 hours of *in vitro* gas production.**
(DOCX)

## Acknowledgments

The authors gratefully acknowledge Universiti Malaysia Sabah (Malaysia) and Universiti Putra Malaysia (Malaysia) for their supports in providing research facilities, and Kuala Lumpur Kepong Berhad (Lahad Datu, Sabah, Malaysia) in providing the shredded OPEFB used in this study.

## Author Contributions

**Conceptualization:** F. M. Y. Nur-Nazratul, M. R. M. Rakib, M. Z. Zailan.

**Data curation:** F. M. Y. Nur-Nazratul.

**Formal analysis:** F. M. Y. Nur-Nazratul.

**Funding acquisition:** M. R. M. Rakib, M. Z. Zailan.

**Investigation:** F. M. Y. Nur-Nazratul.

**Methodology:** F. M. Y. Nur-Nazratul, M. R. M. Rakib, M. Z. Zailan, H. Yaakub.

**Project administration:** M. R. M. Rakib.

**Resources:** M. R. M. Rakib, M. Z. Zailan, H. Yaakub.

**Supervision:** M. R. M. Rakib, M. Z. Zailan, H. Yaakub.

**Visualization:** F. M. Y. Nur-Nazratul, M. R. M. Rakib.

**Writing – original draft:** F. M. Y. Nur-Nazratul.

**Writing – review & editing:** M. R. M. Rakib, M. Z. Zailan.

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
