## [Decision Letter · Decision Letter 0]

14 Jun 2021

PONE-D-21-15493

Enhancing in vitro ruminal digestibility of oil palm empty fruit bunch by biological pre-treatment with Ganoderma lucidum fungal culture

PLOS ONE

Dear Dr. Rakib,

Thank you for submitting your manuscript to PLOS ONE. After careful consideration, we feel that it has merit but does not fully meet PLOS ONE’s publication criteria as it currently stands. Therefore, we invite you to submit a revised version of the manuscript that addresses the points raised during the review process.

We look forward to receiving your revised manuscript.

Kind regards,

Juan J Loor

Academic Editor

PLOS ONE

Journal Requirements:

1. Please ensure that your manuscript meets PLOS ONE's style requirements, including those for file naming. The PLOS ONE style templates can be found athttps://journals.plos.org/plosone/s/file?id=wjVg/PLOSOne_formatting_sample_main_body.pdf and https://journals.plos.org/plosone/s/file?id=ba62/PLOSOne_formatting_sample_title_authors_affiliations.pdf

Additional Editor Comments (if provided):

Reviewers' comments:

Reviewer's Responses to Questions

**Comments to the Author**

1. Is the manuscript technically sound, and do the data support the conclusions?

Reviewer #1: Partly

Reviewer #2: Yes

2. Has the statistical analysis been performed appropriately and rigorously? 

Reviewer #1: Yes

Reviewer #2: Yes

3. Have the authors made all data underlying the findings in their manuscript fully available?

Reviewer #1: Yes

Reviewer #2: Yes

4. Is the manuscript presented in an intelligible fashion and written in standard English?

Reviewer #1: Yes

Reviewer #2: No

5. Review Comments to the Author

Reviewer #1: The work was not well set for objectives to be investigated.

Many research work in this area has been reported but the authors have not made sufficient reviews with relevanct data.

No new innovations have been found and reported under this experiment.

Both the Abstract and Conclusion were not well wrapped up using the significant findings under this experiment.

In particular, comparison of the treated OPEFB after various periods should have been stated comparatively among the treatments.

The authors did not show any suggestions or recommendations to further implement the treated OPEFB in the possible feeding interventions.

Reviewer #2: The study investigated the enhancing of the nutritive value of oil palm empty fruit bunch after a fungal culture treatment. This is a comprehensive and very interesting study for increasing the nutritive value after reduce the lignocellulose content by fungal degradation. The study was well-conducted with reasonable replicates.

I have a question. Because Reishi is used in traditional medicine in the East, its price is high. Would it be profitable to use it to improve the nutritional value of fruit and vegetable by-products? that is, is the process very expensive? Well, when a fruit and vegetable by-product is used, it is to replace a more expensive food. And if the treatment of the by-product with Reishi is expensive, then the cost of the ration will not be lowered. have you evaluated these aspects? Could Reishi affect palatability?

I have few major issues

L10-L13-L153-L155-L157 …. Universiti? Check English language, please.

L77 new paragraph, so tabulate

L88 After Ganoderna lucidum add (G. Lucidum)

L170 NEWAY is any Technology, Company, please clarify

L185 Do you use DMRT abbreviation though the manuscript? If not, you do not need it.

L189 SAS is Statistical Analysis System

L198 I am sorry I could not find what is viz.?

L303-312 correlation is R or r? Please revise the authors guidelines

As I have commented previously, the study is very interesting and the description of the in vitro techniques for the production of gas and the degradability of the dry matter are adequate. But since the main objective is to use this waste byproduct in the feeding of ruminant animals, I miss the determination of some parameters and indices related to ruminal fermentation. I believe that to continue with the publication of this manuscript it should be essential to incorporate at least one table with the pH, ammoniacal nitrogen, total volatile fatty acids, and some individual VFA such us acetate, butyrate, propionate, isobutyrate, valerate and isovalerate. And ideally, it would be possible to have some odd-chain volatile fatty acid. But I think that at least these basic ones that I have indicated must be included. It is probable that the authors have already analyzed them and can incorporate them.

The results and the technology are very interesting. English grammar and expression should be improved.

6. PLOS authors have the option to publish the peer review history of their article (what does this mean?). If published, this will include your full peer review and any attached files.

Reviewer #1: No

Reviewer #2: No

---

## [Author Response · Author response to Decision Letter 0]

16 Jun 2021

The authors would like to thank the reviewers for their specific and helpful comments and suggestions. Please find enclosed the revised manuscript with marked changes (Revised Manuscript with Track Changes) and unmarked (Manuscript). Detail response to the reviewers can be find in 'Response to Reviewers'.

Below, are the general response by the authors:

Respond to editor comments: Contribution statements were added on the title page. The authors also checked throughout the manuscript to ensure the style and formatting. File naming followed the guide given. All figures were uploaded to the Preflight Analysis and Conversion Engine (PACE).

Respond to Reviewer 1 comments: The comments given by Reviewer 1 are very general, and not pointing to any specific content of the manuscript. The authors believed that the study was well set, the changes in fiber composition and in vitro rumen digestibility were well described among the treatments, and new interesting innovation was reported in the study as supported by Reviewer 2. The abstract was improved, and the conclusion already summarized the findings. The authors already stated few limitations, recommendations for future research, and potential applications of the treated OPEFB in the final paragraph of the results and discussion section.

Respond to Reviewer 2 comments: The authors would like to appreciate the merits given to the study and manuscript. The comments given by Reviewer 2 are specific and relevant to the study. The authors addressed most of the issues, and made necessary modification in the manuscript. The use of G. lucidum in this study was in a form of mycelial culture which is easy to multiply/culture, and not the basidiocarp (fruiting body) which is expensive. All minor errors in the manuscript as pointed by Reviewer 2 were corrected. Unfortunately, rumen fermentation characteristics data as suggested by Reviewer 2 was not recorded during the study, and the authors are unable to provide the data due to unavailability of samples to redo the in vitro experiment. The authors feel that, ruminal fermentation characteristics should be evaluated in future study for the final feed product (after the treated OPEFB is mixed with other ingredients to formulate a balanced ratio). The present study focused on improving the digestibility of OPEFB. The authors also believed that we provided sufficient data to support that the treatment of G. lucidum on OPEFB was able to reduce the fibre composition (especially lignin), and enhanced the in vitro rumen digestibility.

---

## [Decision Letter · Decision Letter 1]

1 Sep 2021

PONE-D-21-15493R1

Enhancing in vitro ruminal digestibility of oil palm empty fruit bunch by biological pre-treatment with Ganoderma lucidum fungal culture

PLOS ONE

Dear Dr. Rakib,

Thank you for submitting your manuscript to PLOS ONE. After careful consideration, we feel that it has merit but does not fully meet PLOS ONE’s publication criteria as it currently stands. Therefore, we invite you to submit a revised version of the manuscript that addresses the points raised during the review process.

A FEW REVISIONS ARE STILL NEEDED.

We look forward to receiving your revised manuscript.

Kind regards,

Juan J Loor

Academic Editor

PLOS ONE

Journal Requirements:

Reviewers' comments:

Reviewer's Responses to Questions

**Comments to the Author**

1. If the authors have adequately addressed your comments raised in a previous round of review and you feel that this manuscript is now acceptable for publication, you may indicate that here to bypass the “Comments to the Author” section, enter your conflict of interest statement in the “Confidential to Editor” section, and submit your "Accept" recommendation.

Reviewer #1: (No Response)

Reviewer #2: All comments have been addressed

2. Is the manuscript technically sound, and do the data support the conclusions?

Reviewer #1: Partly

Reviewer #2: Yes

3. Has the statistical analysis been performed appropriately and rigorously? 

Reviewer #1: Yes

Reviewer #2: Yes

4. Have the authors made all data underlying the findings in their manuscript fully available?

Reviewer #1: Yes

Reviewer #2: Yes

5. Is the manuscript presented in an intelligible fashion and written in standard English?

Reviewer #1: Yes

Reviewer #2: Yes

6. Review Comments to the Author

Reviewer #1: Although, the authors have addressed partly to the issues raised, some additional clarifications and modifications are still required;

Line; 370-373, based on the improved nutritional value of the OPEFB by G. lucidum, the authors need to address the improvement and express the lacking parts ..and propose the potential improvement. It would be beneficial to refer to previous workers-References.

Conclusions; it looks as if, the authors still discussing, rather than summarizing the significant findings under this experiment, in particular to the improvement rendered by G. lucidum, to what extent?

Reviewer #2: The manuscript is now ready for publication. The authors have responded to all the questions that were raised and have completed and improved the document.

7. PLOS authors have the option to publish the peer review history of their article (what does this mean?). If published, this will include your full peer review and any attached files.

Reviewer #1: **Yes: **Metha Wanapat

Reviewer #2: No

---

## [Author Response · Author response to Decision Letter 1]

7 Sep 2021

The authors would like to thank the reviewers for their specific and helpful comments and suggestions. Please find enclosed the revised manuscript with marked changes (Revised Manuscript with Track Changes) and unmarked (Manuscript).

Respond to Journal Requirements: The citations and references list were cross-checked, and endured complete and correct. Reference No. 23 was substituted with other reference, as the original reference was unpublished (preprints).

Respond to Reviewer #1: The lacking of the G. lucidum-treated OPEFB was stated, where it contain low amount of crude protein, lipids, vitamins, and minerals. Therefore, it is necessary to incorporate the treated OPEFB with other nutrients as supplements for formulation of a balanced ration. Two additional references were added ([49,50]). The conclusions section was improved. The extent of changes in the G. lucidum treated OPEFB were specified in the conclusion section, the conclusion summarized the major findings, and answered the objectives of the study.

Respond to Reviewer #2: The authors would like to thank Reviewer #2 for his/her kind contribution in reviewing this manuscript.

Other issues: Previously, the rate stated was based on the number of mycelia plugs (5 mm) per 500 g OPEFB. The rate now is expressed as mg/kg of OPEFB.

---

## [Editor Report · Decision Letter 2]

17 Sep 2021

Enhancing in vitro ruminal digestibility of oil palm empty fruit bunch by biological pre-treatment with Ganoderma lucidum fungal culture

PONE-D-21-15493R2

Dear Dr. Rakib,

We’re pleased to inform you that your manuscript has been judged scientifically suitable for publication and will be formally accepted for publication once it meets all outstanding technical requirements.

Kind regards,

Juan J Loor

Academic Editor

PLOS ONE
---

## [Editor Report · Acceptance letter]

21 Sep 2021

PONE-D-21-15493R2 

Enhancing in vitro ruminal digestibility of oil palm empty fruit bunch by biological pre-treatment with *Ganoderma lucidum* fungal culture 

Dear Dr. Rakib:

I'm pleased to inform you that your manuscript has been deemed suitable for publication in PLOS ONE. Congratulations! Your manuscript is now with our production department. 

Kind regards, 

on behalf of

Dr. Juan J Loor 

Academic Editor

PLOS ONE